# miRNAs: Targets to Investigate Herpesvirus Infection Associated with Neurological Disorders

**DOI:** 10.3390/ijms242115876

**Published:** 2023-11-01

**Authors:** Vanessa Cristine de Souza Carneiro, Luciane Almeida Amado Leon, Vanessa Salete de Paula

**Affiliations:** 1Laboratory of Molecular Virology and Parasitology, Oswaldo Cruz Institute, Fiocruz, Rio de Janeiro 21040-360, Brazil; vanessacarneiro@aluno.fiocruz.br (V.C.d.S.C.); vdepaula@ioc.fiocruz.br (V.S.d.P.); 2Laboratory of Technological Development in Virology, Oswaldo Cruz Institute, Fiocruz, Rio de Janeiro 21040-360, Brazil

**Keywords:** microRNAs, herpesvirus, neurological diseases

## Abstract

Herpesvirus is associated with various neurological disorders and a specific diagnosis is associated with a better prognosis. MicroRNAs (miRNAs) are potential diagnostic and prognostic biomarkers of neurological diseases triggered by herpetic infection. In this review, we discuss miRNAs that have been associated with neurological disorders related to the action of herpesviruses. Human miRNAs and herpesvirus-encoded miRNAs were listed and discussed. This review article will be valuable in stimulating the search for new diagnostic and prognosis alternatives and understanding the role of these miRNAs in neurological diseases triggered by herpesviruses.

## 1. Introduction

Human herpesviruses (HHV) have been known to cause skin lesions since ancient Greece [1]. However, advances in research have revealed the pathogenesis of viruses better, and herpesviruses are known to be involved in several other pathologies, including neurological diseases [2]. Herpesviruses can establish latency and be reactivated by host immune stimuli [3,4]. The permanence of herpesviruses in host organisms allows them to contribute to the pathogenesis of various diseases [5,6,7].

There are nine HHV species: human alphaherpesvirus 1 (herpes simplex virus (HSV)-1), human alphaherpesvirus 2 (HSV-2), human alphaherpesvirus 3 (varicella-zoster virus (VZV)-3), human gammaherpesvirus 4 (Epstein–Barr virus, EBV), human betaherpesvirus 5 (human cytomegalovirus, HCMV), human betaherpesvirus 6A, 6B, and 7 (HHV-6A, HHV-6B, and HHV-7), and human gammaherpesvirus 8 (Kaposi’s sarcoma-associated herpesvirus (KSHV)) [8]. Except for KSHV, all herpesviruses can invade the nervous system and trigger pathologies such as encephalitis, meningitis, myelitis, mental confusion, and epilepsy [7,9,10]. However, proper diagnosis can positively influence the clinical outcome owing to the availability of antiviral treatments for herpesvirus infection [11]. However, this diagnosis is often invasive and difficult to perform or interpret, preventing suitable treatment [11,12]. Therefore, investigations into biomarkers for neurological diseases are growing [13,14,15]. Recently, our group identified differentially expressed microRNAs (miRNAs) in a cohort of patients with COVID-19, who had HHV-6 detection and neurological symptoms [16].

miRNAs are small RNAs that act as posttranscriptional regulators involved in various cellular processes [17]. With the involvement of miRNA expression in some pathologies being elucidated, their role as potential biomarkers has also been corroborated [18]. Therefore, these molecules are investigated as potential diagnostic and therapeutic tools [19]. Although studies using miRNAs as diagnostic markers are relatively recent, tests that employ miRNA profiles to diagnose certain types of tumors, such as thyroid nodules, are currently available. Nodules can be classified using miRNAs [20,21]. Further, the potential of miRNAs as biomarkers of neuroinflammation has been indicated because of their involvement in the inflammatory response associated with neurological disorders [22,23,24]. Some miRNAs can promote or suppress inflammatory responses. In addition, they participate in intercellular communications [25]. Such functions indicate that the differential expression of miRNAs is associated with neurological pathologies, and therefore, these miRNAs could function as biomarkers [23]. Thus, the investigation of miRNAs as biomarkers is a promising strategy for diagnostic tests. This review article aims to present the miRNAs associated with CNS and herpes to stimulate the investigation of these biomarkers in the context of neurological disorders associated with this infection.

## 2. miRNAs and Neurological Diseases

miRNA studies have evolved since 1993, when miRNAs were first discovered in *Caenorhabditis elegans* [26,27]. As of August 2023, the number of results for miRNA searches in PubMed is 125,447. Currently, the involvement of miRNAs in several pathologies is known; however, some studies have proposed that miRNAs play crucial roles in the nervous system [23,28,29]. Certain miRNAs have been shown to regulate synaptic function, axon formation, and some other neural functions [30,31]. Some miRNAs have also been reported to be associated with neuroinflammation, controlling the initiation and maintenance of inflammation [23]. For example, miR-21 and miR-155 are involved in the regulation of inflammation through Toll-like receptor (TLR) signaling [23,32,33,34] (Figure 1).

miRNAs regulate neuroinflammation in various neurological diseases and are considered regulators of neuroinflammation resulting from viral infections [25,35,36,37]. Therefore, miRNAs have been associated with several neurological diseases (Table 1). These examples show that the involvement of miRNAs in neurological diseases is well documented. 

## 3. Human and Herpesvirus-Encoded miRNAs

Herpesviruses evade the host immune response and establish permanent infection [49]. One strategy used by some viruses, including herpesviruses, is to regulate the expression of their host or human miRNAs to promote viral infection [50]. Godshalk et al. [51] demonstrated the ability of EBV to modulate the expression of human miRNAs. miR-146a is modulated during EBV infection. It has been hypothesized that the virus encodes or activates a host-specific miRNA suppressor. Furthermore, miR-23a has been observed to contribute to HSV-1 replication by blocking the interferon pathway in the interferon regulatory factor 1 (IRF1), which is involved in innate antiviral immunity [52]. Therefore, the regulation of miRNA expression is assumed to affect other signaling pathways and cellular functions, as human miRNAs can have several mRNA targets and miRNAs have strong regulatory properties to maintain their homeostasis and function [53,54]. Some human miRNAs influenced by the action of herpesviruses are associated with neurological diseases [54]. In a literature search, we identified several human miRNAs associated with neurological diseases with altered expression during herpesvirus infection.

### 3.1. miR-138

The establishment of latency is a mechanism to escape the immune response. To achieve this, herpesviruses use several mechanisms, including using the host’s miRNAs to their advantage [55]. During replication, HSV-1-encoded infected-cell polypeptide 0 (ICP0), a protein common to alphaherpesviruses that regulates lytic and latent infection and mediates their functions by influencing various cellular pathways and proteins involved in cellular defenses, is expressed, leading to the restriction of viral infection [56]. miR-138, a neuron-specific miRNA, targets ICP0 mRNA, promoting the survival of the virus through latency and repression of lytic infection in neurons [57,58,59]. Therefore, miR-138, an abundant miRNA in neurons, is a neuronal factor that contributes to HSV-1 latency [57]. However, the neurological effects of miR-138 regulation during HSV infection are unclear. miR-138 has been studied in neuroscience and is associated with human memory performance and Alzheimer’s disease. The result showed that miR-138 interacts with single nucleotide polymorphisms (SNPs) in genes that can affect human memory performance [60,61].

### 3.2. miR-124

Circulating miR-124 has been proposed as a biomarker of neurological disorders, such as acute ischemic stroke and MS [62,63]. miR-124 is the most abundant miRNA in the brain; in general, its expression has been shown to promote cell differentiation and repress cell proliferation. Furthermore, miR-124 acts as a key regulator of microglial quiescence in the CNS and as a modulator of monocyte and macrophage activation [64]. However, herpesvirus infection also appears to modulate the expression of miR-124. A study investigating whether latent HCMV infection alters the expression of host miRNA showed that miR-124-3p was significantly up-regulated in the HCMV latent infection library, suggesting that this miRNA may be related to the maintenance of latency [65]. However, an involvement of the differential expression of miR-124-3p during HCMV latency with neurological disorders caused by the herpes virus is not known [65]. A study aimed at developing a neuronal cell culture model found that overexpression of miR124 could maintain HSV-1-infected neural cells in quiescent infection, with the accumulation of latency-associated transcript (LAT). Thus, it is hypothesized that miR-124 is involved in neuronal differentiation and thus supports quiescent HSV-1 infection in neurons. It is likely that HSV-1 gene expression can be directly disrupted by miR124, leading to the establishment of latency [66]. However, more studies are needed to investigate the regulatory effect of miR124 on the establishment of latency and, in addition, these findings suggest an investigation into the role of this abundant miRNA in the brain in neurological outcomes caused by herpetic infection.

### 3.3. miR-146a

miR-146a is an important epigenetic modulator of inflammatory signaling and innate immune responses in several neurological disorders, such as Alzheimer’s disease (AD) [67]. In addition to its association with the neuroinflammatory response, hsa-miRNA-146a-5p is significantly up-regulated by various neurotropic DNA viruses, including HSV-1 [68]. HSV-1 infections in neural cells induce miR-146a up-regulation, which is associated with pro-inflammatory signaling in brain cells [69]. Recently, the relationship between HSV-1 infection and miR-146a expression in AD was investigated. A prominent role for miR-146a, which is induced by HSV-1, has been indicated in the activation of key elements of the arachidonic acid cascade and pro-inflammatory pathways, which are known to contribute to neuropathological changes such as AD [70]. It is still unclear whether the significant induction of miRNA-146a after viral infection is a protective mechanism of the cell or a strategy used by the virus for invasion and replication; however, all types of viral infections that induce miRNA-146a expression are associated with neurological diseases [71].

### 3.4. miR-155

A study with microRNA-155 knockout mice (miR-155KO) concluded that miR-155 could result in greater susceptibility of the nervous system to HSV infection [72]. HSV-1 infection up-regulates miR-155-5p, which in turn increases HSV-1 replication by promoting transcription of serine/arginine rich splicing factors (2SRSF2), an important transcriptional activator of viral gene expression [73]. 

miR-155 has also been proposed to act as a liaison between HHV-6 and AD. The role of HHV-6 in the pathology of AD has also been a subject of investigation [58,59]; studies have shown that miR-155 dysregulation can modulate important processes in AD pathogenesis. In mouse models, HHV-6 has been shown to suppress the expression of miR-155 [74,75,76,77]. A study on patients with acute encephalopathy with reduced subcortical diffusion (AED) and HHV-6 detected miR-155 in the CSF, although the difference was not statistically significant [78]. AED is a neurological complication that leads to neurodevelopmental sequelae in children. AED may be caused by HHV-6 [79]. These findings indicate a relationship between AED and miR-155 and reinforce the investigation of miR-155 as a biomarker for neurological diseases caused by herpesviruses.

### 3.5. miR-132

miR-132 is associated with neuronal development and functioning [80]. miR-132 is up-regulated in patients with VZV infection. Based on target gene prediction results, the analysis revealed that the genes targeted by miRNAs are involved in the nervous system, suggesting that miR-132 is associated with VZV-induced nervous system complications [81]. However, careful clinical validation is required to confirm these findings [82].

### 3.6. miR-21

miR-21 is involved in the regulation of inflammatory processes in the nervous system, and this miRNA has been reported to be down-regulated during HCMV infection, demonstrating how HCMV utilizes cellular miRNAs for viral replication [83,84]. HCMV in neural cells has been proposed to inhibit miR-21 to increase the levels of cell cycle regulators and miR-21 targets, which may benefit viral replication. These data provide insights into the investigation of this miRNA in therapeutic interventions, as miR-21 may be involved in viral replication, which may be associated with neuronal damage triggered by the action of the virus [85].

### 3.7. miR-122

This miRNA plays a suppressive role in CNS tumors, and EBV infection reduces the expression of miR-122, which can lead to tumor progression. Thus, it has been proposed that EBV can alter the condition of cancer cells in the CNS by altering the expression of miR-122 and is therefore related to the development of cancer cells [86]. Although this requires further clarification, the involvement of EBV in both neurological diseases and malignancies corroborates these data [87].

### 3.8. miR-Let-7a and miR-Let-7b

The let-7 family of miRNAs is associated with neurodegeneration and neuroinflammation [88]. Similar to those in EBV, miRNAs from the let-7 family are associated with multiple sclerosis (MS) [10,89]. It has been postulated that the EBV protein EBNA-1 transactivates the expression of primary let-7a transcripts. Thus, the up-regulation of miR-let-7a is mediated by EBNA1 [90]. These data allowed us to hypothesize a possible relationship between miR-let-7a, EBV, and neurodegenerative diseases.

An investigation of the interaction between MS risk genes and miRNAs reported that miR-let-7b-5p interacts with the MS risk gene *ZC3HAV1* in EBV-infected B cells. Furthermore, the down-regulation of miR-let-7b-5p in EBV-infected B cells compared to that in uninfected B cells provides evidence that EBV infection down-regulates miR-let-7b-5p, including MS-risk miRNAs, which may contribute to MS pathogenesis via direct or indirect interaction with risk genes [91]. In a study conducted by our group, we found a significant increase in miR-let-7b-5p expression in a group of patients with HHV-6 infection and neurological manifestations, which encouraged studies investigating the role of this miRNA in neurological complications caused by HHV-6 [16].

### 3.9. miR-142-3p

miR-142 has already been shown to be related to gammaherpesvirus infection, being negatively regulated in EBV-positive lymphomas, in addition to having a viral ortholog, kshv-miR-K12-10, which presented sequences similar to miR-142-3p [55,92]. However, miR-142-3p has also been linked to neurological disorders, promoting IL-1β-dependent glutamatergic synaptic dysfunction of the glial glutamate-aspartate transporter (GLAST) [93]. miR-142-3p has been identified as a marker of negative prognosis in patients with MS and suggested as a therapy strategy to improve the course of the disease in multiple sclerosis, a disease that is also closely related to the gammaherpesvirus EBV. Maintaining miR-142-3p at a low level would help improve the prognosis of the disease [94]. Studies indicate that miR-142 is involved in the regulation of lymphocyte Tregs, and the overexpression of miR-142 prevents proper differentiation of Tregs [89]. These findings suggest that increased expression of miR-142 isoforms (miR-142-3p and miR-142-5p) may be involved in the pathogenesis of autoimmune neuroinflammation, such as MS, influencing T cell differentiation, and this effect may be mediated by the interaction of miR-142 isoforms with SOCS1 and transforming growth factor receptor beta 1 (TGFBR-1) transcripts [95].

## 4. Herpesvirus-Derived miRNAs (v-miRs)

In 2004, a study reported virus-derived miRNAs (v-miRs) in EBV-infected cells [96]. Since then, the role of these v-miRNAs has been scrutinized for elucidation. Viruses were demonstrated to encode v-miRs using canonical or non-canonical miRNA biogenesis pathways [96], which may circumvent immune responses and extend the longevity of infected cells. Currently, the data of approximately 1300 mature v-miRs are available in the VIR-miRNA database [97,98]. Recently, the SARS-CoV-2 was also found to encode v-miRNAs that may be associated with neurological disorders [99,100,101]. However, most identified v-miRNAs are encoded and expressed by herpesviruses [55,80]. Studies seeking to clarify the role of these viral miRNAs point to their role in maintaining latency, preventing lytic replication, and allowing viral DNA to remain in the host organism [55,102]. The v-miRNAs found in the three herpesvirus subfamilies (alphaherpesvirus, betaherpesvirus, and gammaherpesvirus) associated with neurological disorders are discussed below.

### 4.1. Alphaherpesvirus v-miRNAs

To date, HSV-1 is known to encode 27 mature miRNA sequences, while HSV-2 is known to encode 24 mature miRNA sequences [103]. Although the functions of HSV-encoded v-miRNAs have not been fully elucidated, v-miRNAs are known to be involved in the regulation of viral latency [104]. HSV is the only member in the subfamily Alphaherpesvirinae to encode v-miRNAs. To date, no miRNAs have been described for VZV [105].

#### 4.1.1. miR-H4-3p

Studies have reported that miR-H4, encoded by HSV-1, targets the viral protein ICP34.5, a key factor in viral neurovirulence required for efficient in vivo viral replication in neurons [106]. miRNA-H4 inhibits ICP34.5 expression to protect latently infected neurons [107]. This information correlates with the results of another study that demonstrated that miR-H4-3p expression was significantly higher in CSF isolated from patients with herpes simplex encephalitis (HSE) than in those isolated from HSV-negative individuals. However, the role of these and other viral miRNAs and their involvement in the neurovirulence of herpesviruses require further clarification [108]. 

#### 4.1.2. miR-H1

As already described, HSV-1 has been constantly associated with Alzheimer’s disease and one of the hypotheses is that the viral infection induces the progression of the disease through the accumulation of beta-amyloid proteins and phosphorylated tau protein [109]. Investigation of the function of the HSV-1-encoded miRNA, miR-H1, revealed that this miRNA is involved in the silencing of ubiquitin protein ligase E3 component n-recognin 1 (Ubr1). Ubr1 has already been identified as responsible for the degradation of fragments of proteins associated with neurodegeneration, such as β-amyloid and phosphorylated tau [110]. The analysis of the role of miR-H1 during HSV-1 replication and its influence on the regulation of Ubr1 revealed that during HSV-1 replication, miR-H1 was widely expressed and negatively regulated the Ubr1 expression, which may explain why Aβ accumulation occurs after the completion of the HSV-1 replication process. In this study, it is suggested that the silencing of Ubr1 may be a mechanism that induces the accumulation of neuronal Aβ by HSV-1 infection, mediated by miR-H1 encoded by HSV-1. Ubr1 plays an important role in the negative regulation of protein fragments that are associated with neurodegenerative disorders [111]. These findings suggest a novel link between HSV-1 and Alzheimer’s disease through v-miRNAs.

#### 4.1.3. miR-H2-3p

Another miRNA encoded by HSV-1 is miR-H2, more predominantly expressed during latency, which suggests that this miRNA may repress the expression of important HSV-1 activators during lytic infection [112]. Alphaherpesviruses establish latency in neurons and, therefore, miR-H2 would favor the maintenance of the virus in neuronal cells [113]. miR-H2-3p, is transcribed in antisense orientation to HSV-1-encoded infected-cell polypeptide 0 (ICP0)—an immediate early protein (IE) that has an E3 ubiquitin ligase activity, highly expressed during lytic infection. ICP0 functions as a viral transcriptional activator important for productive HSV-1 replication and is believed to have a role in reactivation from latency. Therefore, HSV-1-encoded miR-H2-3p targets the viral ICP0 mRNA to regulate viral latency and virulence [114]. In light of such clarifications on the role of miR-H2-3p in maintaining HSV-1 infection in neurons, a study examined the presence of HSV-encoded and expressed miRNAs in herpes simplex encephalitis (HSE) [108]. HSV is the leading cause of viral encephalitis, and has a mortality rate of nearly 30% if left untreated [2]. It was demonstrated that miR-H2-3p was significantly more expressed in CSF-derived exosomes isolated from HSE patients than those isolated from HSV-negative individuals. Therefore, miR-H2-3p may inhibit ICP0 protein expression, thereby decreasing HSV entry into the productive replication cycle, which can be considered a potential mechanism for HSE [108]. These findings encourage investigations into v-miRNAs in the pathogenesis of neurological diseases and their potential application in the clinic of these diseases.

### 4.2. Betaherpesvirus v-miRNAs

HCMV encodes 26 v-miRNAs, HHV-6B encodes 8 mature v-miRNAs, and HHV-6A encodes miR-U86 [103,115,116]. However, reports on HHV-6 v-miRNAs are more recent, and there are few studies on the role of these v-miRNAs. However, several studies have been conducted on HCMV v-miRNAs and their influence on the host. HCMV v-miRNAs target multiple host genes involved in the immune response and cell cycle control [117]. To date, there have been no reports of miRNAs encoded by HHV-7 [104].

#### 4.2.1. miR-aU14 

miR-aU14 is expressed during HHV-6 reactivation [118]. It has also been found in the spinal cord axons of patients with myalgic encephalomyelitis/chronic fatigue syndrome (ME/CFS), a complex multisystem disorder characterized by neurological, metabolic, and immune dysfunction. Although further studies are needed, these findings may indicate a role for this miRNA in the pathogenesis of ME/CFS [119].

#### 4.2.2. hcmv-miR-UL112-3p, hcmv-miR-US25-1-5p and miR-US25-2-5p

The expression of HCMV v-miRNAs, namely hcmv-miR-UL112-3p, hcmv-miR-US25-1-5p, and hcmv-miR-US25-2-5p, was investigated in patients less than 6 months of age with clinical features of congenital CMV infection [120]. It is reported that hcmv-miR-UL112-3p, hcmv-miR-US25-1-5p and miR-US25-2-5p are involved in congenital HCMV infection [121]. Although the authors did not find a significant difference, miR-US25-1-5p and miR-US25-2-5p levels were higher in some newborns with abnormal brain images than in those with normal brain images, indicating the need for further investigation into the role of these miRNAs in neurological manifestations triggered by HCMV [120]. Furthermore, miR-UL112-3p is significantly up-regulated in patients with glioblastoma, a type of brain tumor [122]. miR-UL112 has also been associated with the silencing of the MHC class I-related gene B (MICB) molecule, a critical factor for natural killer (NK) cell function during viral infection favoring viral evasion [123]. This same mechanism has already been seen in miRNAs encoded by other human herpesviruses and associated with the progression of neurological diseases [124].

### 4.3. Gammaherpesvirus v-miRNAs

Most EBV miRNAs are transcribed from the BART and BHRF1 regions. There are 44 mature EBV-encoded miRNAs, whereas KSHV encodes 25 mature miRNAs [103,117]. Similar to other miRNAs encoded by HHVs, these v-miRNAs are also associated with the maintenance of latency and interference with the host immune system [117]. The Gammaherpesvirinae subfamily members encode several miRNAs, indicating their prospects for investigating their pathogenesis. KSHV can also modulate cellular gene expression through miRNA mimicry to exploit cellular miRNA target networks. Through these mechanisms, KSHV miRNAs promote immune evasion, cell survival, and tumorigenesis [90].

#### miR-BART2-5p and miR-BHRF1-3 

EBV encodes several viral miRNAs that induce immune escape, inhibit apoptosis, and cause neuroinflammation [87]. miR-BART2-5p is encoded by EBV, which down-regulates the MICB. MICB binds to the NKG2D receptor, a dominant natural killer (NK) cell activator; dysregulation of NK cells is in turn associated with the pathogenesis of MS [124,125]. A significant increase in the expression of miR-BART2-5p and miR-BHRF1-3 has been observed in the circulation of patients with MS [126]. Furthermore, miR-BART2-5p and miR-BHRF1-3 were also found to be significantly elevated in the plasma of patients with brain tumors [87]. These findings show that EBV miRNAs can lead to immunological imbalances and should therefore be investigated in the context of neurological diseases.

Although discoveries about v-miRNAs are still considered recent, here we present at least four herpesviruses that encode miRNAs involved in neurological factors. These data show that herpesvirus-encoded v-miNAs may have an association with neurological disorders during herpetic infection (Figure 2). Therefore, the investigation of v-miRNAs can lead to the discovery of new biomarkers.

## 5. miRNAs in Neurodegenerative Diseases and Association with Herpesvirus

With recent advances, miRNAs has been widely investigated in the neuropathology of neurodegenerative diseases and proposed as biomarkers [127]. Recently, it was reported that miR-7-1-5p and miR-223-3p together with an increased concentration of circulating α-synuclein may be useful biomarkers in Parkinson’s disease [128]. A report also brought interesting data on miRNAs as possible circulating biomarkers in patients with amyotrophic lateral sclerosis (ALS). This investigation showed a positive and negative regulation of miR-23c and miR-192-5p, respectively, in patients with ALS. The analysis of bioinformatics revealed that these miRNAs interact with different target genes and are involved in the biological processes of ALS [129]. Regarding Alzheimer’s disease, miR-146a is constantly investigated. It is hypothesized that in Alzheimer’s disease, neuronal cells produce more miR-146a, which decreases the levels of protein kinase 1 (ROCK1) and reduces the levels of phosphatase and tensin homolog phosphorylated (p-PTEN), preventing the dephosphorylation of the tau protein (p-tau). Thus, p-tau accumulates in neurons to form neurofibrillary tangles (NFT), ultimately leading to neuronal death in Alzheimer’s disease [130]. Interestingly, neurodegenerative diseases are also constantly associated with the action of herpesvirus infection, such as Alzheimer’s disease, multiple sclerosis, Parkinson’s disease, and ALS [131,132]. miRNAs have been proposed as a link between herpetic infection and some the progression of neurodegenerative diseases. Recent studies have proposed that EBV is the cause of MS, as the risk of MS is seen to increase in individuals with EBV infection [133]. Surprisingly, it was observed that one of the miRNAs encoded by EBV targets genes related to MS, ebv-miR-BHRF1-2-5p directly targeting MALT1, which are involved in the activation of the NF-κB pathway, and regulation of T cells, which may point to the crucial molecular processes in the pathogenesis of MS [134]. Furthermore, HSV-1 is suspected of inducing neurodegeneration in Alzheimer’s disease [128]. One link may be miR-146a, a miRNA commonly associated with Alzheimer’s disease, which has been seen up-regulated during HSV-1 infection. This regulation is associated with pro-inflammatory signaling in brain cells in Alzheimer’s disease [69]. These studies may provide insights into the link between neurodegenerative diseases and herpesviruses.

## 6. Neurological Disorders Caused by Herpesviruses and Their Association with miRNAs

Among the nine herpesviruses, eight are considered neurotropic and are associated with different neurological outcomes [135], except for KSHV, which has not been studied in detail in the context of neurological diseases. Alphaherpesviruses can infect neurons and establish latency. The spread of the virus to the sensory and autonomic nerves creates a reservoir of the virus in the trigeminal or dorsal root ganglia [113,136]. When reactivated, these viruses can bypass the blood–brain barrier and gain access to the CNS, triggering neurological diseases, such as encephalitis, meningitis, postherpetic neuralgia, and vasculopathy [137,138,139,140]. Betaherpesviruses can infect a variety of cell types; however, myeloid lineage cells, especially monocytes, have been identified as the main sites of latency [141,142]. Betaherpesviruses can also infect cells present in the nervous system. Passage across the blood–brain barrier is believed to be mediated by monocytes [143]. Betaherpesviruses can trigger encephalitis, neurodevelopmental deficits, meningoencephalitis, dizziness, and epilepsy [135]. Gammaherpesviruses predominantly establish latency in B cells [144]. However, EBV has been reported to replicate in the CNS and disrupt the integrity of the blood–brain barrier, triggering neurological disorders such as meningitis, encephalitis, myelitis, psychosis, and “Alice in Wonderland” syndrome [10,135]. 

Given the involvement of herpesviruses in neurological diseases, it would be interesting to identify biomarkers of neurological diseases in herpes infections. miRNAs are considered key mediators of the host response to infection, predominantly by regulating proteins involved in immune pathways, and have therefore been proposed as potential biomarkers of neurological diseases [19,23]. Thus, some miRNAs are associated with neurological disorders caused by herpesviruses (Table 2).

The clinical approach to neurological disorders triggered by herpetic infection initially consists of analyzing the clinical signs and symptoms. Imaging exams, such as tomography and resonance, allow to visualize inflammation in the temporal lobe and can be used to help the diagnosis [145]. The quantitative polymerase chain reaction (qPCR) is used to detect the virus at early stages, the detecting viral DNA in cerebrospinal fluid (CSF) is considered the gold standard [135,146]. However, the sensitivity of the test depends on the material collection period, and false negative results may exist when samples are collected too early or too late [147]. One of the explanations for false negative results at the beginning of the infection would be a very low viral load [148]. However, early diagnosis is essential because delay in treatment is associated with a negative prognosis [149]. Therefore, investigating these miRNAs as biomarkers could be an interesting strategy when neurological clinical signs and symptoms are observed.

Although there are many studies that investigate miRNAs as biomarkers of neurological diseases, few of them are dedicated to an association between these miRNAs and neurological diseases caused by herpesviruses and, so far, none of them were used as treatment targeting those miRNAs. Investigations of this type would provide new means of diagnosis and even possible treatments targeting these miRNAs. This is an approach being investigated for the treatment of some neurological disorders such as Alzheimer’s disease, cerebral ischemia, and epilepsy, where the regulation of miRNAs may reduce neurological damage in these pathologies [38,41,150,151].

## 7. Conclusions

miRNAs are secreted into extracellular fluids and can be considered signaling molecules and proposed as biomarkers for various diseases [152]. This review may help and encourage the search for biomarkers of neurological diseases associated with herpesviruses. Human miRNAs are associated with neurological disorders, including herpetic infection [68,72]. In addition, we highlight the importance of investigating v-miRNAs as biomarkers of neurological disorders associated with herpesviruses, as these viruses are capable of coding their own miRNAs as escape routes for the host immune system [107,120]. Thus, we conclude that miRNAs may represent future alternative biomarkers for neurological diseases caused by herpesviruses. This topic is extremely relevant as the prevalence of herpesviruses is high in the general population and these viruses can be reactivated during co-infections and may be associated with neurological disorders, as seen during the COVID-19 pandemic [153]. However, although there is specific treatment for neuroinfections caused by herpesviruses, early diagnosis with specific biomarkers is an essential tool for a good prognosis.

## Figures and Tables

**Figure 1 ijms-24-15876-f001:**
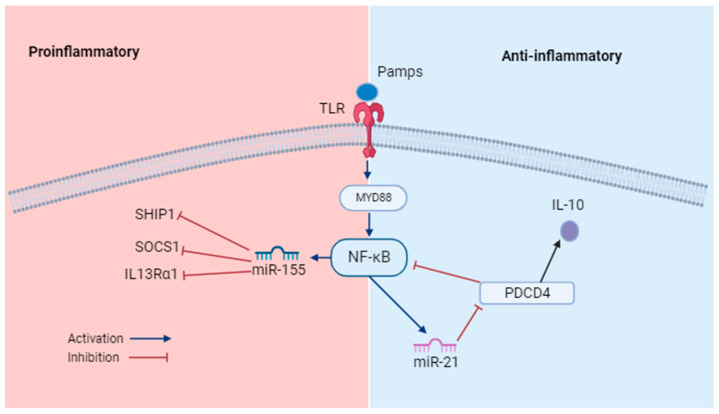
miRNAs and inflammatory signaling. miR-155 promotes inflammation. Its action is induced by nuclear factor κB (NF-κB), after TLR signaling. The increase in the expression of miR-155 inhibits the expression of anti-inflammatory regulators, such as the suppressor of cytokine signaling (SOCS1), a negative regulator of cytokines; SH2 domain-containing Inositol 5’-Phosphatase1 (SHIP1), a negative regulator of TNF-α; and IL-13 alpha 1 receptor (IL13Rα1). miR-21 is induced by TLR4-MyD88-NFκB, the increase in the expression of miR-21 inhibits PDCD4, which participates in pro-inflammatory signaling, which triggers an increase in the expression of Il-10, an anti-inflammatory cytokine.

**Figure 2 ijms-24-15876-f002:**
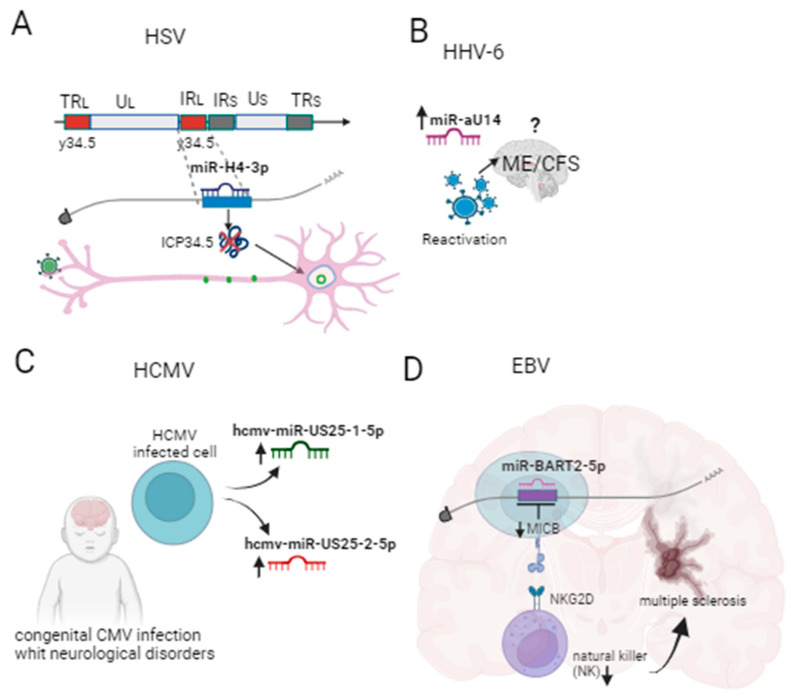
(**A**) Increased miRNA-H4 would be associated with inhibition of ICP34.5 expression, a key factor in HSV-1 neurovirulence, which protects latently infected neurons. (**B**) miR-aU14 would be associated with myalgic encephalomyelitis/chronic fatigue syndrome (ME/CFS). (**C**) hcmv-miR-US25-2-5p and hcmv-miR-US25-1-5p would be associated with congenital CMV infection. (**D**) miR-BART2-5p negatively regulates MICB, a molecule that binds to the NKG2D receptor, an activator of natural killer (NK) cells. The NK cell dysregulation has been linked to MS progression. ^?^ Under investigation.

**Table 1 ijms-24-15876-t001:** miRNAs associated with neurological disorders.

miRNA	Role	Neurological Disorders	Reference
miR-132	Regulator of pro-epileptogenic factors	Epilepsy	[38]
	Regulates synaptic function, axon, dendritic and spinal maturation	Alzheimer’s disease	[39,40,41]
miR-134	Associated with the development of dendritic and synaptic spine	Acute ischemic stroke	[42,43]
miR-124	Regulator of the immune response during neuroinflammation	Ischemic stroke	[44]
miR-223	Regulates myeloid cell and granulocyte differentiation, and dendritic cell activation	Dementia and neurodegenerative diseases, such as multiple sclerosis (MS) and Parkinson’s disease	[45,46]
mIR-137	Influences the expression of many genes implicated in neurodevelopment	Neuropsychiatric disorders	[47,48]

**Table 2 ijms-24-15876-t002:** Viral and human miRNAs associated with neurological disorders and herpes infection.

Human miRNAs			
miRNAs	Subfamily	Virus	Reference
miR-138	Alphaherpesvirinae	HSV-1 and HSV-2	[57,58,59]
miR-124-3p	Alphaherpesvirinae	HSV-1	[66]
miR-146a	Alphaherpesvirinae	HSV-1	[69,70]
miR-155	Alphaherpesvirinae	HSV-1	[72,73]
	Betaherpesvirinae	HHV-6	[74,75,78]
miR-132	Alphaherpesvirinae	VZV	[81]
miR-21	Betaherpesvirinae	HCMV	[85]
miR-122	Gammaherpesvirinae	EBV	[86]
miR-let-7a	Gammaherpesvirinae	EBV	[89,90]
miR-let-7b	Gammaherpesvirinae	EBV	[91]
miR-142	Gammaherpesvirinae	EBV	[89,92,93,94,95]
**Viral miRNAs**			
**miRNAs**	**Subfamily**	**Virus**	**Reference**
miR-H4-3p	Alphaherpesvirinae	HSV-1	[106,107,108]
miR-H1	Alphaherpesvirinae	HSV-1	[97]
miR-H2-3p	Alphaherpesvirinae	HSV-1	[108]
miR-aU14	Betaherpesvirinae	HHV-6	[119]
miR-UL112-3p	Betaherpesvirinae	HCMV	[121,122]
miR-US25-1-5p and miR-US25-2-5p	Betaherpesvirinae	HCMV	[120,121]
miR-BART2-5p and miR-BHRF1-3	Gammaherpesvirinae	EBV	[87,124,125,126]

## Data Availability

Data sharing is not applicable to this article.

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
