# Peer review of "miRNAs: Targets to Investigate Herpesvirus Infection Associated with Neurological Disorders"

_ijms, 2023, doi:10.3390/ijms242115876_

Round 1
Reviewer 1 Report
Comments and Suggestions for Authors
Overall, this paper provides a review about the role of miRNAs in neurological disorders caused by herpesvirus infection. This paper collects the evidence of the relationship between miRNAs and neuro-disorders, which is informative. This work is scientifically interesting and worthy of publication in IJMS. However, the following minor concerns should be addressed before publication. In addition, the editing of the manuscript should be improved.
1. The whole review paper has only one figure, which is not sufficient. First, the authors should highly summarize the relationship between miRNA and neurological disorders. For example, instead of reading the complex information in section 2 (miRNAs and neurological disease), a figure would be much better. It can be similar to Table 1, but it should show the mechanisms about how the miRNAs affect human neuro cells (what pathways?). Please take other review papers as references.
2. There should be a comprehensive discussion on the diagnostics and treatment. As mentioned by the authors, miRNAs should be great biomarkers for neurological disorders caused by herpesvirus infection. What is the clinical approach for diagnostics? Is there novel methods to practically/accurately detect them at early stages? Furthermore, is there any treatment targeting those miRNAs? The authors should have a discussion on these points as perspectives. Instead of just introduce others’ work, the authors opinions can be more valuable.
3. Please prepare high resolution figures (best to prepare vector images not bit maps).
4. Please re-organize the abstract and improve the language, it is the first impression of the whole review paper.
Comments on the Quality of English LanguageModerate editing of English language required.
Author Response
Response to Reviewer 1 Comments
Overall, this paper provides a review about the role of miRNAs in neurological disorders caused by herpesvirus infection. This paper collects the evidence of the relationship between miRNAs and neuro-disorders, which is informative. This work is scientifically interesting and worthy of publication in IJMS. However, the following minor concerns should be addressed before publication. In addition, the editing of the manuscript should be improved.
Point 1. The whole review paper has only one figure, which is not sufficient. First, the authors should highly summarize the relationship between miRNA and neurological disorders. For example, instead of reading the complex information in section 2 (miRNAs and neurological disease), a figure would be much better. It can be similar to Table 1, but it should show the mechanisms about how the miRNAs affect human neuro cells (what pathways?). Please take other review papers as references.
Response 1: We appreciate your comments and hope that we understood and addressed the issues raised. We reduced texts with complex information and including a figure (figure 1) illustrating the role of miRNAs in the inflammatory process during the neuroinflammation. Furthermore, we include a table (table1) that shows the role of miRNAs involved in some neurological disorders.
Point 2. There should be a comprehensive discussion on the diagnostics and treatment. As mentioned by the authors, miRNAs should be great biomarkers for neurological disorders caused by herpesvirus infection. What is the clinical approach for diagnostics? Is there novel methods to practically/accurately detect them at early stages? Furthermore, is there any treatment targeting those miRNAs? The authors should have a discussion on these points as perspectives. Instead of just introduce others’ work, the authors opinions can be more valuable.
Response 2. We have added the information suggested by the reviewer (line 404-423):
“The clinical approach to neurological disorders triggered by herpetic infection initially consists of analyzing the clinical signs and symptoms. Imaging exams, such as tomography and resonance, allow to visualize inflammation in the temporal lobe and can be used to help the diagnosis [146]. The quantitative PolymeraseChain Reaction (qPCR) is used to detect the virus at early stages, the detecting viral DNA in cerebrospinal fluid (CSF) is considered the gold standard. [136,147]. However, the sensitivity of the test depends on the material collection period, and false negative results may exist when samples are collected too early or too late [148]. One of the explanations for false negative results at the beginning of the infection would be a very low viral load [149]. However, early diagnosis is essential because delay in treatment is associated with a negative prognosis [150] . Therefore, investigating these miRNAs as biomarkers could be an interesting strategy when neurogical clinical signs and symptoms are observed.
Although there are many studies that investigate miRNAs as biomarkers of neurological diseases, few of them are dedicated to an association between these miRNAs and neurological diseases caused by herpesviruses and until now any of them was used as treatment targeting those miRNAs. Investigations of this type would provide new means of diagnosis and even possible treatments targeting these miRNAs. This is an approach being investigated for the treatment of some neurological disorders such as Alzheimer's disease, cerebral ischemia, and epilepsy, where the regulation of miRNAs may reduce neurological damage in these pathologies.[38,41,151,152].
Point 3. Please prepare high resolution figures (best to prepare vector images not bit maps).
Response 3. Thank for observation. Figures have been modified according to IJMS standards.
Point 4. Please re-organize the abstract and improve the language, it is the first impression of the whole review paper.
Response 4. Thanks for pointing out this, the summary has been adjusted
Reviewer 2 Report
Comments and Suggestions for Authors
The review article by de Souza Carneiro is a well-rounded and referenced overview of miRNAs as an intersection of herpes virus biology and neurodegenerative diseases. The aims and methodology are clearly defined and the literature review/research very detailed.
The most important aspects of miRNA biology and herpesviridae are mentioned. The review is logically and systematically well-constructed.
Style and grammer are appropriate.
I have only two suggestions which will help to improve the study:
- Include more extensive sections/information on specifically Alzheimer's Disease, Multiple Sclerosis, Parkinson's disease and Amyotrophic Lateral sclerosis. All of the four diseases have been previously well linked to miRNAs in serum, CSF, brain tissue, animal and cell-culture models. The interest in orthoherpesviridae role in these diseases has been rapidly growing recently, consider e.g. Bjornevik et al. 2022 DOI: 10.1126/science.abj8222 or Blackhurst et al. 2023 https://doi.org/10.1038/s41582-023-00790-6.
- I would add miR-142 to the list of reviewed human miRNAs, it has been linked on multiple occasions to immunomodulatory functions, PD moncytes, Alzheimer's disease, etc, multiple sclerosis etc. and I think to orthoherpesviridae, e.g. here: Piedade et al. 2016 doi: 10.3390/v8060156. Consider also additional miRNAs which could have been overlooked.
Additinal minor comments:
"response by binding to receptor tools" -> "to receptors"; + add citation for this statement
"miR-138 has been studied in neuroscience, and associated with human memory performance and Alzheimer disease [56,57]." -> And the results/conclustion was...?
- Line335: I think it's not precise to state that KSHV "has not been studied in detail", while in fact there is a lot of research material on it. Maybe the authors should specify which aspect of KSHV has not been studied in detail?
Author Response
Response to Reviewer 2 Comments
The review article by de Souza Carneiro is a well-rounded and referenced overview of miRNAs as an intersection of herpes virus biology and neurodegenerative diseases. The aims and methodology are clearly defined and the literature review/research very detailed.
The most important aspects of miRNA biology and herpesviridae are mentioned. The review is logically and systematically well-constructed.
Style and grammer are appropriate.
I have only two suggestions which will help to improve the study:
Point 1. Include more extensive sections/information on specifically Alzheimer's Disease, Multiple Sclerosis, Parkinson's disease and Amyotrophic Lateral sclerosis. All of the four diseases have been previously well linked to miRNAs in serum, CSF, brain tissue, animal and cell-culture models. The interest in orthoherpesviridae role in these diseases has been rapidly growing recently, consider e.g. Bjornevik et al. 2022 DOI: 10.1126/science.abj8222 or Blackhurst et al. 2023 https://doi.org/10.1038/s41582-023-00790-6.
Response 1. We greatly appreciate your contributions. A section on miRNAs in neurodegenerative diseases and association with herpesvirus has been included (line 350-378):
“5. miRNAs in neurodegenerative diseases and association with herpesvirus.
With recent advances miRNAs has been widely investigated in the neuropathology of neurodegenerative diseases and proposed as biomarkers [129]. Recently, it was reported that miR-7-1-5p and miR-223-3p together with an increased concentration of circulating α-synuclein may be useful biomarkers in Parkinson's disease [130]. A report also brought interesting data on miRNAs as possible circulating biomarkers in patients with amyotrophic lateral sclerosis (ALS). This investigation showed a positive and negative regulation of miR-23c and miR-192-5p, respectively, in patients with ALS. The analysis of bioinformatics revealed that these miRNAs interact with different target genes and are involved in the biological processes of ALS [131]. Regarding Alzheimer's disease, miR-146a is constantly investigated. It is hypothesized that in Alzheimer's disease, neuronal cells produce more miR-146a, which decreases the levels of protein kinase 1 (ROCK1) and reduces the levels of phosphatase and tensin homolog phosphorylated (p-PTEN), preventing the dephosphorylation of the tau protein (p-tau). Thus, p-tau accumulates in neurons to form neurofibrillary tangles (NFT), ultimately leading to neuronal death in Alzheimer's disease.[132]. Interestingly, neurodegenerative diseases are also constantly associated with the action of herpesvirus infection, such as Alzheimer's disease, multiple sclerosis, Parkinson's disease, and ALS [133,134]. miRNAs have been proposed as a link between herpetic infection and some the progression of neurodegenerative diseases. Recent studies have proposed that EBV is the cause of MS, as the risk of MS is seen to increase in individuals with EBV infection [135]. Surprisingly, it was observed that one of the miRNAs encoded by EBV targets genes related to MS, ebv-miR-BHRF1-2-5p directly targeting MALT1, which are involved in the activation of the NF-κB pathway, and regulation of T cells, which may point to the crucial molecular processes in the pathogenesis of MS [136]. Furthermore, HSV-1 is suspected of inducing neurodegeneration in Alzheimer's disease [129]. One link may be miR-146a, a miRNA commonly associated with Alzheimer's disease, which has been seen upregulated during HSV-1 infection. This regulation is associated with pro-inflammatory signaling in brain cells in Alzheimer's disease.[69]. These studies may provide insights into the link between neurodegenerative diseases and herpesviruses”
Point 2. I would add miR-142 to the list of reviewed human miRNAs, it has been linked on multiple occasions to immunomodulatory functions, PD monocytes, Alzheimer's disease, etc, multiple sclerosis etc. and I think to orthoherpesviridae, e.g. here: Piedade et al. 2016 doi: 10.3390/v8060156. Consider also additional miRNAs which could have been overlooked.
Response 2. Thank you for your suggestion, we have included miR-142 in our list (line 207-222)
3.9. miR-142-3p
miR-142 has already been shown to be related to gammaherpesvirus infection, being negatively regulated in EBV-positive lymphomas, in addition to having a viral ortholog, kshv-miR-K12-10, which presented sequences similar to miR-142-3p[55,92]. However, miR-142-3p has also been linked to neurological disorders, promoting IL-1β-dependent glutamatergic synaptic dysfunction of the glial glutamate-aspartate transporter (GLAST)[93]. miR-142-3p has been identified as a marker of negative prognosis in patients with MS and suggested as a therapy strategy to improve the course of the disease in multiple sclerosis, a disease that is also closely related to the gammaherpesvirus EBV. Maintaining miR-142-3p at a low level would help improve the prognosis of the disease[94]. Studies indicate that miR-142 is involved in the regulation of lymphocyte T regs and overexpression of miR-142 prevents proper differentiation of Tregs [89]. These findings suggest that increased expression of miR-142 isoforms (miR-142-3p and miR-142-5p) may be involved in the pathogenesis of autoimmune neuroinflammation, such as MS, influencing T cell differentiation, and this effect may be mediated by the interaction of miR-142 isoforms with SOCS1 and transforming growth factor receptor beta 1 (TGFBR-1) transcripts.[95].”
Additinal minor comments:
"response by binding to receptor tools" -> "to receptors"; + add citation for this statement
Response: This sentence was removed and figure 1 was included in place of the explanation, as suggested by reviewer 1.
"miR-138 has been studied in neuroscience, and associated with human memory performance and Alzheimer disease [56,57]." -> And the results/conclustion was...?
Response: The requested information has been included (line 114-116):
“The result showed that miR-138 interacts with single nucleotide polymorphisms (SNPs) of genes that can affect human memory performance. [60,61].”
- Line335: I think it's not precise to state that KSHV "has not been studied in detail", while in fact there is a lot of research material on it. Maybe the authors should specify which aspect of KSHV has not been studied in detail?
Response: We appreciate your observation. The text has been changed (line 381-382) to “KSHV, which has not been studied in detail in the context of neurological diseases”.
Round 2
Reviewer 1 Report
Comments and Suggestions for Authors
The revised the manuscript has already addressed the concerns in the previous review comments. In addition, more details such as figures and tables have been added, which help with the understanding of the role of miRNA in neurological disorders. Section 6 provides more comprehensive discussion about diagnostics and potential treatments. Overall, the revised manuscript is suitable for publication in IJMS.